# Multi-Output Distributional Fairness via Post-Processing

**Gang Li**                                                    *gang-li@tamu.edu*
*Texas A&M University*

**Qihang Lin**                                                 *qihang-lin@uiowa.edu*
*The University of Iowa*

**Ayush Ghosh**                                                *ayushghosh70@gmail.com*
*The University of Iowa*

**Tianbao Yang**                                               *tianbao-yang@tamu.edu*
*Texas A&M University*

**Reviewed on OpenReview:** *https://openreview.net/forum?id=MJOKrHqiV1*

## Abstract

The post-processing approaches are becoming prominent techniques to enhance machine learning models' fairness because of their intuitiveness, low computational cost, and excellent scalability. However, most existing post-processing methods are designed for task-specific fairness measures and are limited to single-output models. In this paper, we introduce a post-processing method for multi-output models, such as the ones used for multi-task/multi-class classification and representation learning, to enhance a model's distributional parity, a task-agnostic fairness measure. Existing methods for achieving distributional parity rely on the (inverse) cumulative density function of a model's output, restricting their applicability to single-output models. Extending previous works, we propose to employ optimal transport mappings to move a model's outputs across different groups towards their empirical Wasserstein barycenter. An approximation technique is applied to reduce the complexity of computing the exact barycenter and a kernel regression method is proposed to extend this process to out-of-sample data. Our empirical studies evaluate the proposed approach against various baselines on multi-task/multi-class classification and representation learning tasks, demonstrating the effectiveness of the proposed approach.[1]

## 1 Introduction

In machine learning, multi-output learning is a broadly defined domain (Xu et al., 2019; Liu et al., 2018), where the goal is to simultaneously predict multiple outputs given an input, such as multi-label classification, multi-class classification, multi-target regression, etc. In contrast to conventional single-output learning like binary classification, multi-output learning is characterized by its multi-variate nature, whose outputs exhibit rich information for further handling. Multi-output learning is important for real-world decision-making where final decisions are made by considering and weighting multiple factors and criteria. For example, when applied to college admission, the predicted multi-outputs can represent a prospective student's likelihoods of accepting the offer, needing financial aid, completing the degree, finding a job at graduation, etc. Those outputs are weighted to guide admission decisions though the weights may vary with colleges, majors and years (Jaschik, 2023).

However, multi-output learning for decision-making faces the challenge of bias and fairness. There is plenty of evidence indicating the discriminatory impact of ML-based decision-making on individuals and groups (O'neil, 2017; Datta et al., 2014; Bolukbasi et al., 2016; Barocas & Selbst, 2016; Raji & Buolamwini,

---

[1]Code is available at: https://github.com/GangLii/TAB

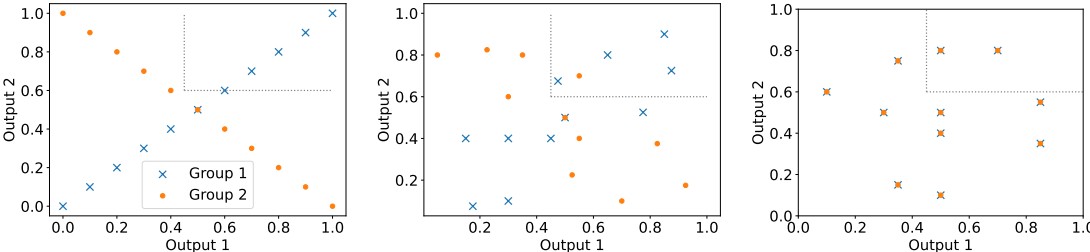

Figure 1: Suppose the original model's outputs (left) minimize prediction error and have a good performance, one can balance performance and fairness by applying our method with setting $\alpha = 0.5$ (middle) or achieve exact fairness by setting $\alpha = 0$ (right).

2019), such as racial bias in assessing the risk of recidivism (Flores et al., 2016) and gender bias in job advertising (Simonite, 2015). To mitigate the bias in machine learning, numerous fairness criteria and algorithms have been proposed (Corbett-Davies et al., 2017; Barocas et al., 2023). These methods introduce statistical constraints during training or post-process the predictions to ensure fair treatment in accordance with corresponding fairness notions such as Demographic Parity (Calders et al., 2009; Chuang & Mroueh, 2021), Equality of Odds or Equal Opportunity (Hardt et al., 2016; Awasthi et al., 2020), Strong Demographic Parity (Agarwal et al., 2019; Jiang et al., 2020) and AUC fairness (Vogel et al., 2021; Yang et al., 2023; Yao et al., 2023). Nevertheless, almost all existing methods focus on ensuring fairness for binary classification or regression within the context of single-output models. Extending fairness to multi-output settings, such as multi-label/multi-class classification and representation learning, remains underexplored and non-trivial.

A naive approach for multi-output fairness is to apply existing fairness-enhancing algorithms for single-output models to each output individually. However, removing unfairness in each output may not help reduce the unfairness in the joint distribution of the outputs. Consider a model with two outputs, with the distributions for two groups shown in Figure 1 (left). The outputs have the same marginal distribution in both groups, so each dimension of the outputs is considered to be fair. However, the outputs have very different joint distributions for different groups, leading to potential unfairness. For example, suppose Output1 and Output2 denote the likelihood of accepting the offer and completing the degree in a college admission procedure, then when a college requires outputs are higher than 0.45 and 0.6 respectively (shown in dot lines in Figure 1), it leads to only students in group 1 being admitted. To address the challenge, in this paper, we propose a post-processing method to enhance fairness of multi-output models on multi-group data by producing a similar distribution of multi-dimensional outputs on each group with minimal manipulation(i.e., minimal mean squared error compared with original outputs). Before discussing details, we present a 2D example in Figure 1.

Measuring fairness based on the distribution of the multi-dimenstional outputs on each group leads to a strong notion called Statistical Parity or Strong Demographic Parity, which has been studied recently for single-output problems in literature (Agarwal et al., 2019; Jiang et al., 2020; Chzhen et al., 2020; Hu et al., 2023). We apply the same notion to multi-output problems and call it Distributional Parity (Definition 1). However, extending this kind of fairness from single-output problems to multi-output problems is non-trivial due to some technical and practical issues: (1) what is the optimal fair predictor for multi-output problems, which achieves Distributional Parity while preserving accuracy best; (2) empirically, how to compute the theoretical optimal fair predictor with only a set of observed data; (3) practically, how to generalize the post-processing method from training data to out-of-sample test data. Our approach addresses the mentioned challenges and our main contributions are summarized below:

- Following Gouic et al. (2020); Chzhen et al. (2020); Chzhen & Schreuder (2022), we derive a post-processing method by projecting a multi-output model to a constraint set defined by a fairness inequality that promotes distributional parity among multiple groups up to a user-specified tolerance of unfairness. We generalize the closed-form solution of the projection by Chzhen et al. (2020); Chzhen & Schreuder (2022) from single-output case to multi-output case using the optimal transport mappings to the Wasserstein barycenter of the model's outputs on each group.

- The aforementioned closed form involves the density function of the model's outputs, which is unknown in practice. Although it can be approximated using training samples, the computation cost of the Wasserstein barycenter is still prohibitively high. To address this issue, we propose to replace the barycenter in the closed form by a low-cost approximate barycenter from Lindheim (2023) using training data. Additionally, we propose a kernel regression approach to extrapolate the closed-form solution, so the post-processing method can be applied to any out-of-sample data.

- The proposed method is task-agnostic and model-agnostic. Numerical experiments are conducted to compare our method with various baselines on multi-label/multi-class classification and representation learning. The results demonstrate the effectiveness of our method.

## 2 Related work

The existing methods to achieve algorithmic fairness include three primary categories: 1) pre-processing methods that exclude sensitive features from data prior to training machine learning models (Dwork et al., 2012; du Pin Calmon et al., 2018; Kamiran & Calders, 2012; d'Alessandro et al., 2017); 2) in-processing methods, which attain fairness during the training phase of the learning model (Agarwal et al., 2018; Goh et al., 2016; Zhang et al., 2018; Kim et al., 2019; Hong & Yang, 2021; Du et al., 2021; Mo et al., 2021; Yao et al., 2023); and 3) post-processing methods designed to alleviate unfairness in the model inferences after the learning process (Hardt et al., 2016; Lipton et al., 2018; Cui et al., 2023; Petersen et al., 2021; Lohia et al., 2019; Xian et al., 2023). The pre-processing methods are inadequate if the sensitive information can be inferred from the remaining variables. The in-processing methods have high computational cost since they typically require re-training a model when the tolerance of unfairness changes. Our method belongs to the last category, which can be directly applicable to any pre-trained model and thus avoids the high computational cost from re-training a model.

This work is motivated by Gouic et al. (2020); Chzhen et al. (2020); Chzhen & Schreuder (2022) where task-agnostic post-processing methods are developed on the empirical cumulative density function and the quantile function of the output, which is limited to single-output models. We extend their approaches for multi-output models using the optimal transport mappings to the barycenter of the model's outputs on different groups. Different from their methods, we approximate the barycenter to reduce the computational cost and apply kernel regression to extend our post-processing method to any new data.

There also exist post-processing methods for specific machine learning tasks. For example, assuming a Bayes optimal score function is available, the group thresholding methods by Gaucher et al. (2023); Chzhen et al. (2019); Schreuder & Chzhen (2021); Zeng et al. (2022) are developed for single-task binary classification. A similar thresholding method has been developed by Denis et al. (2021) for multi-class classification where the thresholds are computed using the Lagrangian multipliers of the demographic parity constraints. The method by Xian et al. (2023) is also for multi-class classification where the output of the score function is mapped a class label by the optimal transport mapping to a Wasserstein barycenter with restricted supports. Compared to these works, the method in this paper is task-agnostic and can be applied to multi-task learning and (self-supervised) representation learning. Hu et al. (2023) developed an approach to enforce strong demographic parity in multi-task learning including both regression and binary classification tasks. Their method essentially applies the single-output method by Gouic et al. (2020); Chzhen et al. (2020) to each task separately, which may not guarantee the distributional parity in this paper. See the example in Figure 1. Though there is a large number of literature working on learning representations independent of sensitive attributes, they achieve this by adversarial training (Zhang et al., 2018; Kim et al., 2019; Xie et al., 2017; Madras et al., 2018), variational auto-encoders (Louizos et al., 2015; Sarhan et al., 2020; Amini et al., 2019; Gong et al., 2024), or contrastive learning(Han et al., 2023; Qi et al., 2024; Zhang et al., 2022; Park et al., 2022), which are mostly in-processing methods and thus have higher computational cost than our method in general.

The multi-output fairness problem in this paper is closely related to compositional fairness introduced in Dwork & Ilvento (2018), which highlights that individually fair classifiers may not compose into fair systems. While Dwork & Ilvento (2018) focuses on categorical outputs, we examine continuous output distributions.

Since categorical outputs often result from thresholding or softmax on continuous outputs, compositional fairness is a special case of multi-output fairness, and our method can directly address it.

## 3 Preliminaries

We consider a general multi-output machine learning problem, e.g., multi-output classification, multi-output regression and representation learning. Let $(X, S)$ be a random vector, where $X \in \mathbb{R}^d$ is the feature vector and $S \in \mathcal{S}$ is a sensitive attribute. It is assumed that $S = [m] := \{1, ..., m\}$. A sample from the distribution of $(X, S)$ is denoted by $(x, s)$. We denote by $f^* : \mathbb{R}^d \times \mathcal{S} \to \mathbb{R}^k$ a machine learning model learned to generate a $k$-dimensional output. For example, $f^*$ can be a multi-task regression model to predict a vector of continuous values, a multi-class/multi-task classification model that returns the probability of each class/task, or a feature extractor that outputs a high-dimensional vector of an input data such as an image.

The focus of this work is not to train the model $f^*$. Assuming $f^*$ is already obtained, this paper studies how to measure and improve its fairness using a post-processing method to modify the output $f^*(X, S)$. More specifically, given $f^*$, we consider a numerical approach for building a mapping $f : \mathbb{R}^d \times \mathcal{S} \to \mathbb{R}^k$ that approximates $f^*$ but produces an output $f(X, S)$ more fair than $f^*(X, S)$. Here, the approximation error is measured by

$$\mathcal{R}(f) = \mathbb{E}\|f^*(X, S) - f(X, S)\|_2^2. \tag{1}$$

If $f^*$ performs well for the task it was built for and the performance metric is continuous, a small $\mathcal{R}(f)$ also ensures $f$ has a reasonably good performance, too.

### 3.1 Distributional Parity

Many existing definitions and measures of fairness in literature are task-specific including, for example, demographic parity, equalized odds and equal opportunity (Dwork et al., 2012; Hardt et al., 2016), which are specific to classification problems. In order to be task-agnostic and model-agnostic, we are interested in the fairness measure that is applied to the distribution of $f(X, S)$ and $f^*(X, S)$. A measure of this kind is the strong demographic parity introduced by Agarwal et al. (2019); Jiang et al. (2020); Chzhen et al. (2020), which essentially indicates that a model $f(x, s)$ is fair only when $f(X, S)$ has the same distribution conditioning on $S = s$ for any $s \in \mathcal{S}$. Although they only consider a single-output model, their fairness measure also applies to the multi-output case. Next we present their fairness measure formally with the name multi-output distributional parity. This new name helps differentiate it from the traditional demographic parity for binary classification (Calders et al., 2009).

**Definition 1** *(Multi-output Distributional Parity) A measurable mapping $f : \mathbb{R}^d \times \mathcal{S} \to \mathbb{R}^k$ satisfies distributional parity if, for every $s, s' \in \mathcal{S}$, $f(X, S)$ conditioning on $S = s$ and $f(X, S)$ conditioning on $S = s'$ have the same distribution.*

**Remark:** Although we focus on Definition 1, an extension of demographic parity, it is natural to extend other notions of fairness, such as equal odds and equal opportunity, in a similar fashion. In particular, suppose $f(X, S)$ is used to predict a target variable $Y$ which takes values from a finite set $\mathcal{Y}$ (e.g., multi-class/mult-label classification). We say $f$ satisfies **distributionally equal opportunity** with respect to a class $y \in \mathcal{Y}$ if, for every $s, s' \in \mathcal{S}$, $f(X, S)$ conditioning on $(S, Y) = (s, y)$ and $f(X, S)$ conditioning on $(S, Y) = (s', y)$ have the same distribution. Similarly, we say $f$ satisfies **distributionally equal odds** if, for every $s, s' \in \mathcal{S}$ and every $y \in \mathcal{Y}$, $f(X, S)$ conditioning on $(S, Y) = (s, y)$ and $f(X, S)$ conditioning on $(S, Y) = (s', y)$ have the same distribution.

Consider a joint distribution of $(X, S, Y)$ where $Y \in \mathbb{R}^p$ is a target vector. In the setting of a single-task binary classification, we have $k = 1$, $p = 1$ and $Y \in \{1, -1\}$ is the true class label. Then the model $f(X, S) \in \mathbb{R}$ typically represents a score/probability of positivity, which is used to predict $Y$ by comparing $f(X, S)$ with a threshold $\theta$, namely, the predicted label $\widehat{Y}$ generated as $\widehat{Y} = 1$ if $f(X, S) \geq \theta$ and $\widehat{Y} = -1$ if $f(X, S) < \theta$. The traditional demographic parity (Calders et al., 2009; Dwork et al., 2012) holds if $\mathbb{P}(\widehat{Y} = 1 | S = s) = \mathbb{P}(\widehat{Y} = 1 | S = s') \quad \forall s, s' \in \mathcal{S}$. Obviously, when $f$ satisfies distributional parity, it satisfies demographic parity for any threshold $\theta$. In fact, this is still true for multi-task binary classification problems, where $p = k > 1$ and $Y \in \{1, -1\}^k$.

The notion of demographic parity has been extended for the multi-class classification problem, where $k > 1$, $p = 1$ and $Y \in [k]$ (Xian et al., 2023). In this case, each coordinate of $f(X, S)$ represents the score of one class in $[k]$ and the predicted label is typically generated as $\widehat{Y} = \arg\max_{l \in [k]} f_l(X, S)$. According to Xian et al. (2023), a model satisfies demographic parity when

$$\mathbb{P}(\widehat{Y} = y | S = s) = \mathbb{P}(\widehat{Y} = y | S = s') \quad \forall s, s' \in \mathcal{S}, \ y \in [k], \tag{2}$$

which is clearly implied by distributional parity.

## 3.2 Wasserstein Distance and Wasserstein Barycenters of Discrete Distributions

It is challenging and, sometimes, unnecessary to obtain a model $f$ that exactly satisfies the distributional parity in Definition 1. In practice, a model slightly violating Definition 1 can be acceptable for some applications. To quantify the extent to which a model $f$ satisfies the distributional parity, a statistical distance is often introduced to measure the difference between the distributions of $f(X, s)$ and $f(X, s')$ for any $s$ and $s'$ in $\mathcal{S}$. Following the literature (Chzhen et al., 2020; Gouic et al., 2020), we utilize the Wasserstein distance that conveys the distinction between probability measures by quantifying the expense associated with transforming one probability measure into another.

**Definition 2** *(Wasserstein-2 distance). Let $\mu$ and $\nu$ be two probability measures on $\mathbb{R}^k$ with finite second moments. The squared Wasserstein-2 distance between $\mu$ and $\nu$ is defined as*

$$\mathcal{W}_2^2(\mu, \nu) = \inf_{\gamma \in \Gamma_{\mu,\nu}} \int_{\mathbb{R}^k \times \mathbb{R}^k} \|x - y\|_2^2 d\gamma(x, y) \tag{3}$$

*where $\Gamma_{\mu,\nu}$ is the set of probability measures on $\mathbb{R}^k \times \mathbb{R}^k$ such that its marginal distributions are equal to $\mu$ and $\nu$, i.e., for all $\gamma \in \Gamma_{\mu,\nu}$ and all measurable sets $A, B \subset \mathbb{R}^k$ it holds that $\gamma(A \times \mathbb{R}^k) = \mu(A)$ and $\gamma(\mathbb{R}^k \times B) = \nu(B)$.*

Suppose probability measures $\mu$ and $\nu$ both have density functions and the infimum in (3) is obtained at $\gamma^*$. There exists a mapping $T_{\mu,\nu} : \mathbb{R}^k \to \mathbb{R}^k$ such that $\gamma^* = (\mathrm{Id}, T_{\mu,\nu}) \# \mu$, where $\#$ denotes the pushforward operator on measures and Id denotes the identity mapping (Santambrogio, 2015). Here, $T_{\mu,\nu}$ is known as the **optimal transport mapping** from $\mu$ to $\nu$. With the Wasserstein distance, we can characterize the geometric average of finitely many probability distributions by the Wasserstein-2 barycenter, which will be later used in the measure of unfairness considered in this paper.

**Definition 3** *For probability measures $\nu_1, \ldots, \nu_m$ and $p_1, \ldots, p_m$ such that $p_i > 0, \sum_{i=1}^m p_i = 1$, the weighted Wasserstein-2 barycenter is given by*

$$\nu^* = \arg\min_\nu \left\{ \Psi(\nu) := \sum_{i=1}^m p_i \mathcal{W}_2^2(\nu_i, \nu) \right\} \tag{4}$$

## 3.3 Post-processing with Distributional Parity Constraint

Let $p_s = \mathbb{P}(S = s)$ and $\nu_{f|s}$ be the probability distribution of $f(X, S)$ conditioning on $S = s$ for $s \in \mathcal{S}$. Following Chzhen & Schreuder (2022), we then measure the unfairness of model $f$ by the sum of the weighted distances between $\nu_{f|s}$ to their weighted Wasserstein-2 barycenters, namely,

$$\mathcal{U}(f) = \min_\nu \sum_{s \in \mathcal{S}} p_s \mathcal{W}_2^2(\nu_{f|s}, \nu). \tag{5}$$

As Wasserstein-2 distance serves as a distance metric on the space of probability distributions, $\mathcal{W}_2^2(\nu_{f|s}, \nu_{f|s'}) = 0$ if and only if $\nu_{f|s} = \nu_{f|s'}$ (Kwegyir-Aggrey et al., 2023; Peyré et al., 2017). Hence, $\mathcal{U}(f) = 0$ if and only if for any $s, s' \in \mathcal{S}, \nu_{f|s} = \nu_{f|s'}$, satisfying the distributional parity. However, when some level of unfairness is allowed, we only need to ensure $\mathcal{U}(f) \leq \alpha \mathcal{U}(f^*)$, where $\alpha \in [0, 1]$ represents the tolerance of unfairness in terms of the fraction of the unfairness of $f^*$. Like Chzhen & Schreuder (2022); Kwegyir-Aggrey et al. (2023), we consider the post-processing problem with distributional parity constraint as follows:

$$\min_f \mathcal{R}(f) \quad \text{s.t.} \quad \mathcal{U}(f) \leq \alpha \mathcal{U}(f^*), \tag{6}$$

where $\mathcal{R}(f)$ is defined in (1).

# 4 Structure of Optimal Solution

Problem (6) has been thoroughly studied by Chzhen et al. (2020); Chzhen & Schreuder (2022) under the setting that $f$ is a single-output model. They provide an intuitive closed form of the optimal solution of (6) for any $\alpha \in [0, 1]$ using the cumulative density function (CDF) of $\nu_{f^*|s}$ for each $s$. As they only focus on a single-output model, we expand their results to the multi-output case with some modifications in their proofs. We present the extension of their results in this section and highlight the main difference.

When $\alpha = 0$ in (6), the optimal solution of (6) can be characterized by the following theorem.

**Theorem 1** *Suppose $\nu_{f^*|s}$ has density and finite second moments for each $s \in \mathcal{S}$. Then*

$$\min_{\mathcal{U}(f)=0} \mathcal{R}(f) = \min_{\nu} \sum_{s \in \mathcal{S}} p_s \mathcal{W}_2^2(\nu_{f^*|s}, \nu) = \mathcal{U}(f^*). \tag{7}$$

*Moreover, if $f_0$ and $\nu_0$ solve the first and second minimization in (7), respectively, then $\nu_0$ is the distribution of $f_0$ and*

$$f_0(x, s) = T_{f^*|s, \nu^0}(f^*(x, s)) \tag{8}$$

*where $T_{f^*|s, \nu_0} : \mathbb{R}^k \to \mathbb{R}^k$ is the optimal transport mapping from $\nu_{f^*|s}$ to $\nu_0$.*

By Definition 3, $\nu_0$ solving the second minimization in (7) is the barycenter of $\{\nu_{f^*|s}\}_{s \in \mathcal{S}}$. This suggests a post-processing method by transporting the output $f^*(X, S)$ when $S = s$ to that barycenter.

When $k = 1$, Theorem 1 is reduced to the structural results obtained by Chzhen et al. (2020); Gouic et al. (2020) (see, e.g., Theorem 2.3 in Chzhen et al. (2020)). Indeed, when $k = 1$,

$$T_{f^*|s, \nu_0}(f^*(x, s)) = \sum_{s' \in S} p_{s'} Q_{f^*|s'} \circ F_{f^*|s}(f^*(x, s)), \tag{9}$$

where $F_{f^*|s}$ is the CDF of $\nu_{f^*|s}$ and $Q_{f^*|s}$ is the quantile function of $\nu_{f^*|s}$, i.e., $Q_{f^*|s}(t) = \inf\{y \in \mathbb{R} : F_{f^*|s}(y) \geq t\}$.

In practice, when some level of unfairness is acceptable, one can select $\alpha > 0$ in (6). When $\alpha = 1$, $f = f^*$ is the optimal solution. When $\alpha \in (0, 1)$, a closed form of the optimal solution for (6) is derived by Chzhen & Schreuder (2022); Kwegyir-Aggrey et al. (2023). Although they are originally studied only under the single-output case, the same result holds for a multi-output model by almost the same proof. This closed form is presented in the following proposition, which motivates the way of trading off the error and fairness in our post-processing method in the next section.

**Proposition 1** *(Proposition 4.1 in Chzhen & Schreuder (2022)) Suppose $\nu_{f^*|s}$ has density and finite second moments for each $s \in \mathcal{S}$. For any $\alpha \in [0, 1]$, the optimal solution to (6), denoted by $f_\alpha$, satisfies (up to a zero-measure set)*

$$f_\alpha(x, s) = \sqrt{\alpha} f^*(x, s) + (1 - \sqrt{\alpha}) f_0(x, s), \tag{10}$$

*where $f_0$ is defined in Eq. (8).*

**Remark:** *It is not always possible to minimize $\mathcal{R}$ and $\mathcal{U}$ simultaneously by the same $f$. Therefore, a model $f$ is valuable as long as it achieves a value of $\mathcal{R}$ (or $\mathcal{U}$) that cannot be further reduced without increasing $\mathcal{U}$ (or $\mathcal{R}$). In fact, a function $f(x, s)$ measurable in $x$ is called Pareto efficient if there is no function $f'(x, s)$ measurable in $x$ such that one of the following two cases happens: (1) $\mathcal{R}(f) \leq R(f')$ and $\mathcal{U}(f) < U(f')$ or (2) $\mathcal{R}(f) < R(f')$ and $\mathcal{U}(f) \leq U(f')$. The Pareto frontier is 2-dimensional curve consisting of $(\mathcal{U}(f), \mathcal{R}(f))$ for all Pareto efficient $f$'s. Proposition 4.6 in Chzhen & Schreuder (2022) shows, the Pareto frontier for $\mathcal{U}$ and $\mathcal{R}$ is actually the curve $\{(\mathcal{R}(f_\alpha), \mathcal{U}(f_\alpha))\}$ with $\alpha$ varying from 0 to 1, where $f_\alpha$ is from (10). Although $d = 1$ in Chzhen & Schreuder (2022), their Proposition 4.6 holds for our setting also for any dimension $d \geq 1$ by the same proof.*

# 5 Fair Post-Processing with Finite Samples

Theorem 1 and Proposition 1 only characterize the optimal solution to (6) when $\nu_{f^*|s}$ has density. However, in practice, we only have access to a finite set of outputs of $f^*$, denoted by $\{f^*(x_i, s_i)\}_{i=1}^n$, where $\{(x_i, s_i)\}_{i=1}^n$

is a dataset sampled from the distribution of $(X, S)$. [2] As a result, we are not able to compute $\nu_0$ and $T_{f^*|s,\nu_0}$ exactly and apply (10). To address this issue when $k = 1$, a plug-in principle is applied by replacing $p_s$, $F_{f^*|s}(t)$, and $Q_{f^*|s}(t)$ in (9) using their empirical approximation based on finite data sample (Chzhen et al., 2020; Gouic et al., 2020). Because $f^*$ is a multi-output mapping in our case, (9) is not applicable. In the following, we present how to employ the plug-in principle by approximating $\nu_0$ and $T_{f^*|s,\nu_0}$ in (8) by finite data sample. It consists of three main steps.

## 5.1 Optimal Transportation with Finite Samples

Consider approximating $\nu_0$ and $T_{f^*|s,\nu_0}$ by a collection of datasets $D_s = \{(x_i^s, s)\}_{i=1}^{n_s}$ for $s \in \mathcal{S}$. We denote the empirical distribution of $f^*$ on $D_s$ by $\nu_{f^*(D_s)}$, i.e., $\nu_{f^*(D_s)} = \frac{1}{n_s} \sum_{i=1}^{n_s} \delta_{f^*(x_i^s, s)}$ for $s \in \mathcal{S}$, where $\delta$ is the Dirac measure. Consider two discrete distributions in $\mathbb{R}^k$: $\mu = \sum_{i=1}^{n_\mu} p_i^\mu \delta_{\xi_i^\mu}$ and $\nu = \sum_{i=1}^{n_\nu} p_i^\nu \delta_{\xi_i^\nu}$, where $\xi_i^\mu \in \mathbb{R}^k$, $\xi_i^\nu \in \mathbb{R}^k$, $p_i^\mu > 0$, $p_i^\nu > 0$, $\sum_{i=1}^{n_\mu} p_i^\mu = 1$ and $\sum_{i=1}^{n_\nu} p_i^\nu = 1$. As a special case of (3), the squared Wasserstein-2 distance between $\mu$ and $\nu$ is

$$\mathcal{W}_2^2(\mu, \nu) = \min_{\gamma \in \mathbb{R}_+^{n_\mu \times n_\nu}} \sum_{i=1}^{n_\mu} \sum_{j=1}^{n_\nu} c_{ij} \gamma_{ij} \quad \text{s.t.} \quad \sum_{i=1}^{n_\nu} \gamma_{ij} = p_i^\mu, \forall j, \quad \sum_{j=1}^{n_\mu} \gamma_{ij} = p_i^\nu, \forall i. \quad (11)$$

where $c_{ij} = \|\xi_i^\mu - \xi_j^\nu\|_2^2$ and $\gamma_{ij}$ represents the mass located $\xi_i^\mu$ transported to $\xi_j^\nu$ in order to move distribution $\mu$ to $\nu$. Also, $\gamma$ can be viewed as a discrete distribution in $\mathbb{R}^k \times \mathbb{R}^k$ supported on $(\xi_i^\mu, \xi_j^\nu)$ for $j = 1, ..., n_\mu$ and $i = 1, ..., n_\nu$. Suppose $\gamma^* \in \mathbb{R}_+^{n_\mu \times n_\nu}$ is the optimal solution of (11). The optimal transportation from $\mu$ to $\nu$ and from $\nu$ to $\mu$, denoted by $T_{\mu,\nu}$ and $T_{\nu,\mu}$ respectively, are random mappings such that

$$\mathbb{P}(T_{\mu,\nu}(\xi_i^\mu) = \xi_j^\nu) = \frac{\gamma_{ij}^*}{p_i^\mu}, \quad \mathbb{P}(T_{\nu,\mu}(\xi_j^\nu) = \xi_i^\mu) = \frac{\gamma_{ij}^*}{p_j^\nu}, \quad (12)$$

for $j = 1, ..., n_\mu$ and $i = 1, ..., n_\nu$.

With $\mathcal{W}_2^2(\mu, \nu)$ given in (11), the barycenter of discrete distributions $\nu_i$ for $i = 1, \dots, m$ is also defined as the solution of (4). Since $\nu_{f^*(D_s)}$ is the discrete approximation of $\nu_{f^*|s}$, we propose to approximate $\nu_0$ in Theorem 1 by the barycenter of $\{\nu_{f^*|s}\}_{s \in \mathcal{S}}$ with the weights $p_s = \frac{n_s}{\sum_{s' \in \mathcal{S}} n_{s'}}$. Unfortunately, Wasserstein barycenters are NP-hard to compute (Altschuler & Boix-Adsera, 2022). Although (4) can be formulated as a multi-marginal optimal transport problem (Agueh & Carlier, 2011) and solved as a linear program (Anderes et al., 2016), its computational complexity scales exponentially in terms of $m$. To reduce the exponential computing complexity, instead of computing the barycenter exactly, we adopt the approach by Lindheim (2023) to construct its approximation. This approach achieves a good balance between runtime and approximation error in practice. We present this approach in the next section.

## 5.2 Approximate Barycenter with Finite Samples

The approach by Lindheim (2023) first computes the optimal transport mapping between $\nu_{f^*(D_s)}$ to $\nu_{f^*(D_s')}$ for each pair of $s$ and $s'$ in $\mathcal{S}$, i.e., $T_{\nu_{f^*(D_s)}, \nu_{f^*(D_{s'})}}$ satisfying (12). For simplicity of notation, we denote $T_{\nu_{f^*(D_s)}, \nu_{f^*(D_{s'})}}$ by $T_{s,s'}$. Then, they define a mapping $M(f^*(x_i^s, s)) = \sum_{s' \in \mathcal{S}} p_{s'} \mathbb{E}T_{s,s'}(f^*(x_i^s, s))$, where $p_s = n_s/(\sum_{s' \in \mathcal{S}} n_{s'})$ and the expectation is taken over the random output of $T_{s,s'}$ following distribution in (12). Here, $M(f^*(x_i^s, s))$ is the weight average of the expected outcomes after transporting $f^*(x_i^s, s)$ to each of the $|\mathcal{S}|$ distributions. Finally, the approximate barycenter is constructed as a discrete distribution with $\sum_{s \in \mathcal{S}} n_s$ supports defined as follows $\tilde{\nu}_0 = \sum_{s \in \mathcal{S}} \sum_{i=1}^{n_s} \frac{p_s}{n_s} \delta_{M(f^*(x_i^s, s))}$. Compared to the exact barycenter, this approximation requires solving $|\mathcal{S}|(|\mathcal{S}| - 1)/2$ optimal transport mapping between two discrete distributions and thus has a polynomial time complexity. This procedure is formally stated in Algorithm 1.

Let $m = |\mathcal{S}|$ and $\nu_s = \nu_{f^*(D_s)}$ in (4). Let $\hat{\nu}_0$ be optimal solution of (4), i.e, the barycenter of $\{\nu_{f^*(D_s)}\}_{s \in \mathcal{S}}$. It is shown by Lindheim (2023) that $\Psi(\tilde{\nu}_0)/\Psi(\hat{\nu}_0) \leq 2$. This bound is significant because there is no

---

[2]The technique we propose is model-agnostic and can be applied directly to the outputs $\{f^*(x_i, s_i)\}_{i=1}^n$, so it does not require knowing $\{(x_i, s_i)\}_{i=1}^n$.

---

**Algorithm 1** Approximate Barycenter

---

1: **Input:** A mapping $f^* : \mathbb{R}^d \times S \to \mathbb{R}^k$ and a dataset $D = \{(x_i, s_i)\}_{i=1}^n$ sampled from the distribution of $(X, S)$.
2: Partition $D$ into subsets based on $s$ and obtain $D_s = \{(x_i^s, s)\}_{i=1}^{n_s}$ for $s \in \mathcal{S}$.
3: Let $p_s = \frac{n_s}{\sum_{s' \in \mathcal{S}} n_{s'}}$ for $s \in \mathcal{S}$.
4: **for** $1 \leq s < s' \leq |\mathcal{S}|$ **do**
5:    Solve (11) with $\mu = \nu_{f^*(D_s)}$ and $\nu = \nu_{f^*(D_{s'})}$ to obtain $T_{s,s'} = T_{\nu_{f^*(D_s)}, \nu_{f^*(D_{s'})}}$.
6: **end for**
7: Construct mapping $M(f^*(x_i^s, s)) = \sum_{s' \in \mathcal{S}} p_{s'} \mathbb{E} T_{s,s'}(f^*(x_i^s, s))$
8: **Output:** Discrete distribution $\tilde{\nu}_0 = \sum_{s \in \mathcal{S}} \sum_{i=1}^{n_s} \frac{p_s}{n_s} \delta_{M(f^*(x_i^s, s))}$.

---

polynomial-time algorithm can achieve a ratio arbitrarily close to one with high probability (Altschuler & Boix-Adsera, 2022).

Although Proposition 1 is derived for continuous distribution, it motivates a heuristic post-processing method to update the output $f^*(x_i^s, s)$ to

$$\tilde{f}_\alpha(x_i^s, s) := \sqrt{\alpha} f^*(x_i^s, s) + (1 - \sqrt{\alpha}) T_{\nu_{f^*(D_s)}, \tilde{\nu}_0}(f^*(x_i^s, s)) \tag{13}$$

for $i = 1, \ldots, n_s$ and $s \in \mathcal{S}$, where $\alpha \in [0, 1]$. Note that $T_{\nu_{f^*(D_s)}, \tilde{\nu}_0}$ is not obtained during Algorithm 1 and needs to be solved separately. When $\alpha = 0$ and $(x_i^s, s)$ is uniformly randomly sampled from $D_s$, $\tilde{f}_\alpha(x_i^s, s)$ has the same distribution for any $s$, indicating the post-processed outputs satisfy distributional parity. However, (13) is only defined for existing data in $D_s$ for $s \in \mathcal{S}$. In the next section, we propose a method to extend this processing scheme to new samples from $(X, S)$.

### 5.3 Post-Process Out-of-Sample Data

Note that $\tilde{f}_\alpha(x_i^s, s)$ is defined for $(x_i^s, s) \in D_s$ only because the optimal transport mapping $T_{\nu_{f^*(D_s)}, \tilde{\nu}_0}$ in (13) is only defined on $f^*(x_i^s, s)$ with $(x_i^s, s) \in D_s$. To extend the definition of $\tilde{f}_\alpha(x_i^s, s)$ from $D_s$ to any $(x, s) \in \mathbb{R}^d \times \mathcal{S}$, we extrapolate $T_{\nu_{f^*(D_s)}, \tilde{\nu}_0}$ over $\mathbb{R}^k$ using the Nadaraya-Watson kernel regression method (Nadaraya, 1964).

Let $K : \mathbb{R}^k \to \mathbb{R}_+$ be a kernel function satisfying $K(z) = K(-z)$ and $\int_{\mathbb{R}^k} K(z) dz = 1$. Let $h > 0$ be a bandwidth. For any $(x, s) \in \mathbb{R}^d \times \mathcal{S}$, the kernel regression estimator of $T_{\nu_{f^*(D_s)}, \tilde{\nu}_0}(f^*(x, s))$ is

$$\bar{f}(x, s) := \tilde{T}_{\nu_{f^*(D_s)}, \tilde{\nu}_0}(f^*(x, s)) := \sum_{i=1}^{n_s} \frac{K\left((f^*(x, s) - f^*(x_i^s, s))/h\right) T_{\nu_{f^*(D_s)}, \tilde{\nu}_0}(f^*(x_i^s, s))}{\sum_{j=1}^{n_s} K\left((f^*(x, s) - f^*(x_j^s, s))/h\right)}. \tag{14}$$

Here, $\bar{f}(x, s)$ is the post-processed prediction for data $(x, s)$. Recall that $T_{\nu_{f^*(D_s)}, \tilde{\nu}_0}$ is a randomly mapping (see (12)), so is $\tilde{T}_{\nu_{f^*(D_s)}, \tilde{\nu}_0}$. To analyze the fairness of $\bar{f}(x, s)$ on the out-of-sample data distribution, we present the following proposition to characterize how much $\bar{f}(X, S)$ violates the distributional parity after post-processing by bounding $\mathcal{U}(\bar{f}(X, S))$ and the Wasserstein distance between the distributions of $\bar{f}(X, S)$ on different groups. The proof is given in Section A.2.

**Proposition 2** *Suppose $K(z) = \frac{1}{\sqrt{2\pi}} \exp(-z^2)$. Conditioning on data $D = \{(x_i, s_i)\}_{i=1}^n$, it holds that*

$$\mathcal{U}(\bar{f}(X, S)) \leq \sum_{s \in \mathcal{S}} O\left(p_s n_s^2 \exp\left(-\frac{1}{h^2}\right)\right) + \sum_{s \in \mathcal{S}} O\left(\frac{p_s}{h^4} \mathcal{W}_2^2(\nu_{f^*|s}, \nu_{f^*(D_s)})\right). \tag{15}$$

*Moreover, let $\nu_{\bar{f}|s}$ be the distribution of $\bar{f}(X, S)$ conditioning on $S = s$ for $s \in \mathcal{S}$. It holds that*

$$\mathcal{W}_2^2(\nu_{\bar{f}|s}, \nu_{\bar{f}|s'}) \leq O\left((n_s^2 + n_{s'}^2) \exp\left(-\frac{1}{h^2}\right)\right) + O\left(\frac{1}{h^4} \mathcal{W}_2^2(\nu_{f^*|s}, \nu_{f^*(D_s)})\right). \tag{16}$$

In both upper bounds above, the first term is the error due to extrapolation which will converge to zero as the bandwidth $h$ approaches zero. The second term is the error due to the sampling error between training data $D$ and the ground-truth data distribution. The second error is represented by the Wasserstein distribution

---

**Algorithm 2** Post-Processing Method by Transporting to Approximate Barycenter (TAB)

---

1: **Input:** A data point $(x, s) \in \mathbb{R}^d \times \mathcal{S}$, a mapping $f^* : \mathbb{R}^d \times S \to \mathbb{R}^k$ and a dataset $D = \{(x_i, s_i)\}_{i=1}^n$ sampled from the distribution of $(X, S)$.
2: Partition $D$ into subsets based on $s$ and obtain $D_s = \{(x_i^s, s)\}_{i=1}^{n_s}$ for $s \in \mathcal{S}$.
3: Compute $\tilde{\nu}_0$ by Algorithm 1.
4: **for** $s = 1, \ldots, |\mathcal{S}|$ **do**
5:     Solve (11) with $\mu = \nu_{f^*(D_s)}$ and $\nu = \tilde{\nu}_0$ to obtain $T_{\nu_{f^*(D_s)}, \tilde{\nu}_0}$.
6: **end for**
7: **if** $(x, s) \in D_s$ **then** compute $\tilde{f}_\alpha(x, s)$ as in (13) **else** compute $\tilde{f}_\alpha(x, s)$ as in (18).
8: **Output:** $\tilde{f}_\alpha(x, s)$

---

$W_2^2(\nu_{f^*|s}, \nu_{f^*(D_s)})$. As the data sizes $n_s$ for $s \in \mathcal{S}$ increase, the rate at which $W_2^2(\nu_{f^*|s}, \nu_{f^*(D_s)})$ converges to zero has been well studied in literature. For example, according to case (3) of Theorem 2 in Fournier & Guillin (2015), if there exists $q > 4$ such that

$$\mathbb{E}_{(X,S)|S=s}\|f^*(X, S)\|_2^q < +\infty,$$

where $\mathbb{E}_{(X,S)|S=s}$ is the expectation over $(X, S)$ conditioning on $S = s$, then the Wasserstein distance in the second term in (15) and (16) satisfies the following concentration inequality

$$\mathbb{P}(W_2^2(\nu_{f^*|s}, \nu_{f^*(D_s)}) \geq \eta) \leq a(n_s, \eta) + b(n_s, \eta) \text{ for any } \eta \in (0, 1) \tag{17}$$

where

$$a(n_s, \eta) = C \begin{cases} \exp(-cn_s\eta^2), & \text{if } k < 4, \\ \exp\left(-cn_s\left(\eta/\log(2 + 1/\eta)\right)^2\right), & \text{if } k = 4, \quad \text{and} \quad b(n_s, \eta) = n_s^{-\frac{q-4}{4}}\eta^{-\frac{q}{4}}. \\ \exp(-cn_s\eta^{k/2}), & \text{if } k > 4. \end{cases}$$

After extrapolating $T_{\nu_{f^*(D_s)}, \tilde{\nu}_0}$ to $\tilde{T}_{\nu_{f^*(D_s)}, \tilde{\nu}_0}$ defined above, we can extend (13) for any out-of-sample data by updating $f^*(x, s)$ to

$$\tilde{f}_\alpha(x, s) := \sqrt{\alpha}f^*(x, s) + (1 - \sqrt{\alpha})\bar{f}(x, s). \tag{18}$$

We then formally present this post-processing method in Algorithm 2. Note that when $(x, s) \in D_s$, we still apply the original mapping in (13) instead of its extrapolation (18).

Our post-processing method can also be extended to improve the fairness of a model under the notion of distributionally equal opportunity or distributionally equal odds introduced in the remark after Definition 1. In particular, to improve the fairness in terms of distributionally equal odds, we can partition the training set $D$ into $|\mathcal{Y}|$ subsets $\{D^y\}_{y \in \mathcal{Y}}$ depending on the class label $Y$. Then we apply Algorithm 2 to each subset $D^y$ to construct a separate post-processing mapping $\bar{f}_y(x, s)$ like (14) for each combination of group $s$ and class $y$. To apply it on an out-of-sample data point, we first generate a predicted label $\hat{y}$ using $f^*(x, s)$ and then convert $f^*(x, s)$ to $\bar{f}_{\hat{y}}(x, s)$ using the mapping constructed on $D^y$. The same procedure works for distributionally equal opportunity with respect to one class $y \in \mathcal{Y}$, except that we only need to construct the mapping on $D^y$ and post-process data for one class $y$.

## 6 Experiments

In this section, we apply the proposed post-processing method to machine learning models with multiple outputs to evaluate its effectiveness, including multi-label/multi-class classification and representation learning.

**Datasets.** In our experiments, we include four datasets from various domains, including marketing domain(*Customer* dataset [3]), medical diagnosis( *Chexpert* Dataset (Irvin et al., 2019)), face recognition (*CelebA* dataset (Liu et al., 2015) and *UTKFace* dataset (Zhang & Qi, 2017)). The details of these datasets are provided in Appendix B.

**Baselines and Settings.** To verify the effectiveness, we compare our method against six baseline approaches, including four post-processing methods: (1)*Hu et al. (2023)* (Hu et al., 2023), which introduces a post-

---

[3]https://www.kaggle.com/datasets/kaushiksuresh147/customer-segmentation

processing approach for incorporating fairness into multi-task learning using multi-marginal Wasserstein barycenters; (2) *Xian et al (2023)* (Xian et al., 2023), which characterizes the inherent tradeoff of demographic parity in multi-class classification and proposes an effective post-processing method; (3) *FRAPPE* (Tifrea et al., 2024), which proposes a generic framework that converts any regularized in-processing method into a post-processing approach; (4) *Adv Debiasing* (Zhang et al., 2018), which presents a framework for mitigating demographic group biases by introducing a group-related variable and jointly learning a predictor and adversary, and two in-processing methods: (5) *SimFair* (Liu et al., 2023), which focuses on training models for fairness-aware multi-label classification; (6) *f-FERM* (Baharlouei et al., 2024), which presents a unified stochastic optimization framework for fair empirical risk minimization regularized by f-divergence measures.

Specifically in our experiments, for baseline *Adv Debiasing*, both Predictor Block and Adversary Block are implemented by a two-layer neural network with 128 hidden units and *tanh* activating function, respectively. For baselines *FRAPPE*, *SimFair*, and *Adv Debiasing*, we train the models for 60 epochs with Adam optimizer, batch size as 64, and tune the learning rate in {1e-3, 1e-4}. We vary their weight parameter $\lambda$(or $\alpha$) in { 0.1, 1, 2, 4, 8, 10} to show the trade-off between classification performance and fairness. For *f-FERM*, we follow their paper to vary the weight parameter $\lambda$ in {0.1, 1, 10, 50, 100, 150} and tune the learning rate in {0.1, 0.01, 0.001} with their proposed optimization algorithm. For *Hu et al. (2023)* (Hu et al., 2023), we vary parameter $\alpha$ for their method in {0, 0.2, 0.4, 0.6, 0.8, 1.0}. Following their paper, the parameter $\alpha$ for Xian et al. (2023) are in {1, 0.16, 0.14, 0.12, 0.1, 0.08, 0.06, 0.04, 0.02, 0.01, 0.008, 0.006, 0.004, 0.002, 0.001, 0.0.}. For our method, we experiment with a Gaussian kernel and $h$ is chosen from {0.02, 0.04, 0.5, 1} based on input dimension, as smaller $h$ theoretically and empirically leads to better performance but too small $h$ may lead to numerical issues. We vary $\alpha$ for our method in {0, 0.2, 0.4, 0.6, 0.8, 1.0}. All the experiments are run five times with different seeds.

## 6.1 Multi-label classification

For multi-label classification tasks, we experiment on *CelebA* dataset and *Chexpert* dataset. We compare our method with *SimFair* (Liu et al., 2023), *FRAPPE* (Tifrea et al., 2024), *Adv Debiasing* and the method *Hu et al. (2023)* (Hu et al., 2023), which essentially independently applies the post-processing method for a single-output model in Chzhen et al. (2020); Chzhen & Schreuder (2022) to each coordinate of the output of a multi-task classification model. Firstly, one ResNet50 (He et al., 2016) and one DenseNet121 (Huang et al., 2017) and are trained on the *CelebA* and *Chexpert* data respectively, then post-processing methods *Hu et al. (2023)* (Hu et al., 2023), *Adv Debiasing* and our proposed method are applied to the predicted probabilities of each task and methods *SimFair* (Liu et al., 2023) and *FRAPPE* (Tifrea et al., 2024) are applied to the extracted image features to train a new linear classification head. For both *FRAPPE* and *SimFair*, the fairness regularizer is defined as described in Equation 1 of the (Liu et al., 2023) paper, as it has been demonstrated to be effective for promoting demographic parity. The results are summarized in Fig. 2(a) and 2(b). We can observe that, at the same level of accuracy, our method achieves much lower unfairness than baselines, which can be seen clearly when drawing horizontal lines across the figures. Particularly, on the Chexpert dataset, our method delivers fairer predictions with only a negligible decrease in predictive performance. From Fig. 2(a) and 2(b), we also note that learning-based baselines such as *FRAPPE*, *SimFair*, and *Adv Debiasing* tend to achieve lower unfairness but only at the cost of a substantial drop in accuracy, whereas processing-based methods like *Hu et al. (2023)* and our method maintain a better balance. Furthermore, we observe that with the same fairness regularizer, the post-processing method *FRAPPE* achieves comparable fairness-error trade-offs to the in-processing technique *SimFair*, aligning with the findings reported in Tifrea et al. (2024).

## 6.2 Multi-class classification

To verify the effectiveness of our proposed method on multi-class classification tasks, we apply our method to the *customer* dataset. A classic multi-class logistic regression model is built first on the training data and then post-processing methods are applied to the predicted probabilities of each class. For in-processing methods, the multi-class logistic regression models are trained with fairness regularizer from scratch. We compare our method with *FRAPPE* (Tifrea et al., 2024), *f-FERM* (Baharlouei et al., 2024), *Xian et al (2023)* (Xian et al., 2023) and *Adv Debiasing*. Note that method *Xian et al (2023)* directly produces the class labels instead of score function, so we can only perform the comparison on the classification accuracy and the unfairness based

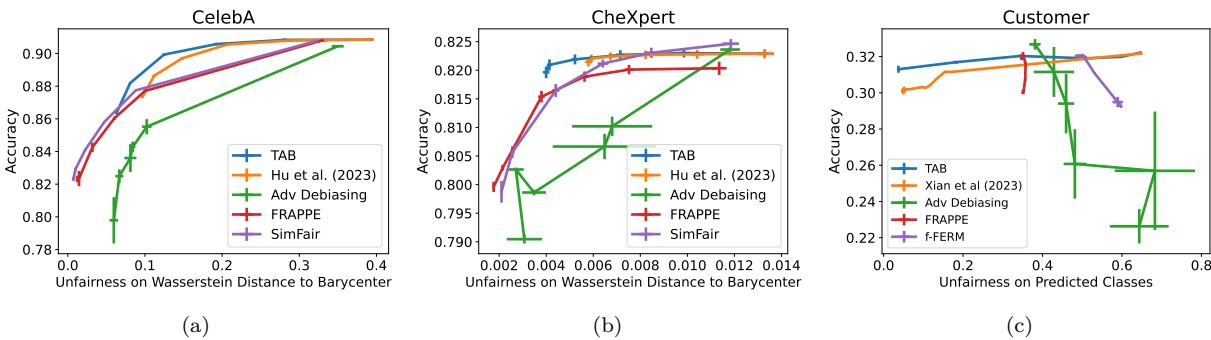

(a)           (b)           (c)

Figure 2: Multi-label classification on CelebA dataset (a) and Chexpert dataset (b); (c) Multi-class classification on Customer dataset.

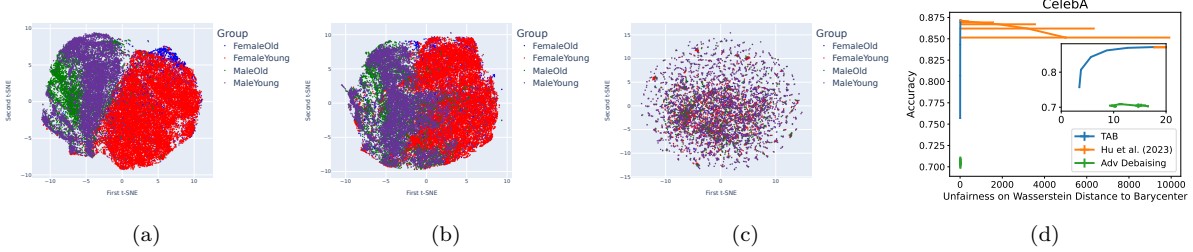

(a)           (b)           (c)           (d)

Figure 3: t-SNE visualization of representations on CelebA dataset. (a) Raw representations from an SSL model; (b) Representations after post-processing($\alpha = 0$) by Hu et al. (2023); (c) Representations after post-processing($\alpha = 0$) with TAB(Ours); (d) Performance on downstream tasks.

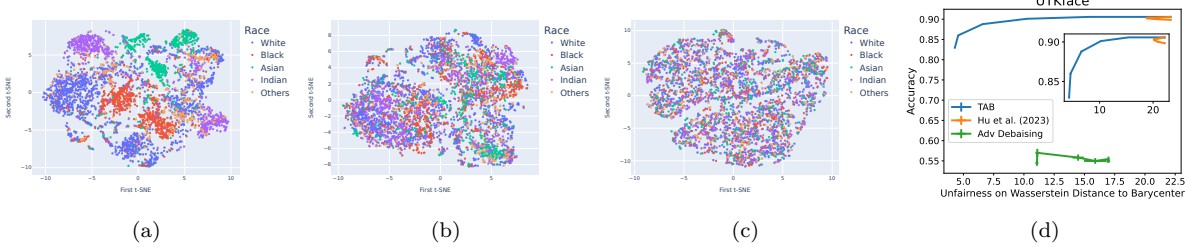

(a)           (b)           (c)           (d)

Figure 4: t-SNE visualization of representations on UTKface Dataset. (a) Raw representations from a CLIP model; (b) Representations after post-processing($\alpha = 0$) with Hu et al. (2023); (c) Representations after post-processing($\alpha = 0$) with TAB(Ours); (d) Performance on downstream tasks.

on Equation 2, which is in favor of their method. For *FRAPPE*, the fairness regularizer is defined as the Equation 2 proposed by Xian et al. (2023). We present the results in Fig. 2(c). The figure demonstrates that our method effectively reduces unfairness measured by the metric proposed by Xian et al. (2023) and achieves a superior balance between accuracy and unfairness compared to the baselines, particularly when a higher level of fairness is required. Besides, we notice that although it is implied that methods *f-FERM* (Baharlouei et al., 2024) and *FRAPPE* (Tifrea et al., 2024) are applicable for multi-class classification, their respective papers do not include experiments for this scenario, and their methods, in our experiments, fail to mitigate unfairness defined by Equation 2 even when it brings a significant undesirable performance decrease.

### 6.3 Representation learning

In this part, we explore the fair representation learning for self-supervised learning(SSL) models and large pre-trained foundation models. We experiment on *CelebA* dataset and *UTKFace* dataset. Due to the scarcity of the postprocessing methods for representation learning in existing literature, we still compare our method with the post-processing method *Hu et al. (2023)* (Hu et al., 2023) and *Adv Debiasing*. For CelebA dataset, we first train an SSL model to learn representations with a dimension of 128, by employing the algorithm proposed in Yuan et al. (2022) on the whole training dataset. Then we apply post-processing methods to eliminate the sensitive information in the raw representations. Specifically, for *Adv Debiasing*, we utilize the raw representations as the labels for the Predictor as there is no task-specific labels in self-supervised learning. In other words, we aim for the model to remove sensitive information while altering the raw representation as minimally as possible. With processed representations, we also build a multi-label logistic regression model to evaluate the performance on downstream tasks. For UTKFace dataset, raw representations are generated by a released CLIP(ViT-B/16) model (Radford et al., 2021) which is pre-trained on 400M text-image pairs, with a dimension of 512.

We evaluate the unfairness of the representations as well as the accuracy performance of the downstream tasks. The results are summarized in Fig. 3 and 4. By examining the representations of Fig. 3(a) and 4(a), it's evident that both the SSL model and CLIP model disclose sensitive attributes prominently. Even after applying post-processing techniques proposed in *Hu et al. (2023)* (Hu et al., 2023), this exposure persists, as depicted in Fig. 3(b) and Fig. 4(b). However, through our proposed methods illustrated in Fig. 3(c) and Fig. 4(c), we are able to achieve fairer representations with respect to sensitive groups. Notably, from Fig. 3(d) and 4(d), we can observe that our method achieves a better tradeoff between downstream performance and distributional parity, whereas *Hu et al. (2023)* (Hu et al., 2023) fails to improve the distributional parity because, as discussed in Fig. 1, the elimination of unfairness in individual outputs may not necessarily mitigate unfairness in the joint distribution of outputs.

## 7 Conclusion and Discussion

In this paper, we have proposed a post-processing method to enhance fairness for multi-output machine learning models, which is underexplored in the literature. Our approach employs optimal transport to move a model's outputs across different groups towards their empirical Wasserstein barycenter to achieve the model's distributional parity. We have developed an approximation technique to reduce the complexity of computing the exact barycenter and a kernel regression method for extending this process to out-of-sample data. Extensive experimental results on multi-label/multi-class classification and representation learning demonstrate the effectiveness of our method.

One limitation of this work is that the notion of fairness in Definition 1 we pursue is strong while a weak notion such as (2) might be sufficient for some specific applications. To achieve a stronger sense of fairness may lead to more decreases in the predictive performance than targeting a weak sense. Therefore, applicability of the proposed method may vary depending on the specific use case. Additionally, the lack of theoretical convergence analysis of the proposed method as the training sample size increases is another limitation, which is an important future work.

### Broader Impact Statement

This paper is to attain distributional fairness in machine learning-driven decision-making, particularly concerning various demographic groups such as gender, age, and race. When enforcing group fairness, it may potentially disadvantage certain individuals. Other than this, none we feel must be specifically highlighted here.

### Acknowledgments

We thank anonymous reviewers for constructive comments. G. Li, Q. Lin, and T. Yang were partially supported by NSF Award 2147253.

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

## A    Proofs

Before providing the proof of Theorem 1, let's review some additional results about Optimal Transport theory.

The next result shows that as long as two measures admit a density with finite second moments, there exists a unique deterministic optimal transport map between them.

**Lemma 1** *(Theorem 1.22 in Santambrogio (2015)) Let $\mu, \nu$ be two measures on $\mathbb{R}^k$ with finite second moments such that $\mu$ has a density and let $X \sim \mu$. Then there exists a unique deterministic mapping $T : \mathbb{R}^k \to \mathbb{R}^k$ such that*

$$\mathcal{W}_2^2(\mu, \nu) = \mathbb{E}\|X - T(X)\|_2^2,$$

*that is $(X, T(X)) \sim \bar{\gamma} \in \Gamma_{\mu,\nu}$ where $\bar{\gamma}$ is an optimal coupling.*

### A.1    Proof of Theorem 1

This proof is originally from the proof of Theorem 2.3 in Chzhen et al. (2020). We extend their results from single output case to the multi-output case with some modifications in their proofs.

**Theorem 1** Suppose $\nu_{f^*|s}$ has density and finite second moments for each $s \in \mathcal{S}$. Then

$$\min_{\mathcal{U}(f)=0} \mathcal{R}(f) = \mathcal{U}(f^*) = \min_{\nu} \sum_{s \in \mathcal{S}} p_s \mathcal{W}_2^2(\nu_{f^*|s}, \nu).$$

Moreover, if $f_0$ and $\nu_0$ solve the first and second minimization in (7), respectively, then $\nu_0$ is the density of $f_0$ and

$$f_0(x, s) = T_{f^*|s, \nu^0}(f^*(x, s))$$

where $T_{f^*|s,\nu_0} : \mathbb{R}^k \to \mathbb{R}^k$ is the optimal transport mapping from $\nu_{f^*|s}$ to $\nu_0$.

**Proof** *First, we'd like to show that*

$$\min_{\mathcal{U}(f)=0} \mathbb{E}\|f^*(X, S) - f(X, S)\|^2 \geq \min_{\nu} \sum_{s \in S} p_s \mathcal{W}_2^2(\nu_{f^*|s}, \nu).$$

*Let $\bar{g} : \mathbb{R}^d \times \mathcal{S} \to \mathbb{R}^k$ be the minimizer of the l.h.s of the above equation and denoted by $\nu_{\bar{g}}$ the distribution of $\bar{g}$. Since $\nu_{f^*|s}$ admits density, with Lemma 1, for each $s \in S$ there exists $T_{f^*|s,\bar{g}}$ such that*

$$\sum_{s \in S} p_s \mathcal{W}_2^2(\nu_{f^*|s}, \nu_{\bar{g}}) = \sum_{s \in S} p_s \int \|z - T_{f^*|s,\bar{g}}(z)\|^2 d\nu_{f^*|s}(z)$$

$$= \sum_{s \in S} p_s \int_{\mathbb{R}^d} \|f^*(x, s) - T_{f^*|s,\bar{g}}(f^*(x, s))\|^2 d\mathbb{P}_{X|S=s}(x)$$

$$= \sum_{s \in S} p_s \mathbb{E}\big[\|f^*(X, s) - T_{f^*|s,\bar{g}}(f^*(X, s))\|^2 | S = s\big]$$

$$= \mathbb{E}\|f^*(X, S) - \tilde{g}(X, S)\|^2$$

*where we define $\tilde{g}(x, s) = T_{f^*|s,\bar{g}}(f^*(x, s))$ for all $(x, s) \in \mathbb{R}^d \times S$. With optimal transportation, $\tilde{g}(X, s)|S = s$ follow the distribution $\nu_{\bar{g}}$ for any $s \in S$. Then we have*

$$\mathcal{U}(\tilde{g}) = \min_{\nu} \sum_{s \in \mathcal{S}} p_s \mathcal{W}_2^2(\nu_{\tilde{g}|s}, \nu) = 0$$

*which indicates $\tilde{g}$ is fair. By optimality of $\bar{g}$ we have*

$$\mathbb{E}\|f^*(X, S) - \tilde{g}(X, S)\|^2 \geq \mathbb{E}\|f^*(X, S) - \bar{g}(X, S)\|^2$$

*Due to definition of $\mathcal{W}_2^2$, for each $s \in S$ we have*

$$\mathcal{W}_2^2(\nu_{f^*|s}, \nu_{\bar{g}}) \leq \mathbb{E}[\|f^*(X, S) - \bar{g}(X, S)\|^2 | S = s]$$

*Then we can conclude*

$$\sum_{s \in S} p_s \mathcal{W}_2^2(\nu_{f^*|s}, \nu_{\bar{g}}) = \min_{\mathcal{U}(f)=0} \mathbb{E}\|f^*(X,S) - f(X,S)\|^2$$

*This implies that*

$$\min_{\mathcal{U}(f)=0} \mathbb{E}\|f^*(X,S) - f(X,S)\|^2 \geq \min_{\nu} \sum_{s \in S} p_s \mathcal{W}_2^2(\nu_{f^*|s}, \nu). \tag{19}$$

*Second, we are going to show that the opposite inequality also holds. To this end, we define $\nu_0$ as*

$$\nu_0 \in \arg\min_{\nu} \sum_{s \in S} p_s \mathcal{W}_2^2(\nu_{f^*|s}, \nu)$$

*Since we assume $\nu_{f^*|s}$ admits density, with Lemma 1, there exists $T_{\nu_{f^*|s}, \nu_0}$ as a optimal transport map from $\nu_{f^*|s}$ to $\nu_0$. And we define $f_0$ for all $(x,s) \in \mathbb{R}^d \times S$ as*

$$f_0(x,s) = T_{\nu_{f^*|s}, \nu_0} \circ f_0(x,s)$$

*By the definition of $f_0$ and the Lemma 1, we have*

$$\min_{\nu} \sum_{s \in S} p_s \mathcal{W}_2^2(\nu_{f^*|s}, \nu) = \mathbb{E}\|f^*(X,S) - f_0(X,S)\|^2 \tag{20}$$

*Moreover since $\nu_0$ is independent from $S$, using similar argument as above we can show that $f_0$ is fair, and it yields*

$$\min_{\nu} \sum_{s \in S} p_s \mathcal{W}_2^2(\nu_{f^*|s}, \nu) \geq \min_{\mathcal{U}(f)=0} \mathbb{E}\|f^*(X,S) - f(X,S)\|^2. \tag{21}$$

*Therefore, combining Eq. 19 and Eq. 21, we showed that*

$$\min_{\nu} \sum_{s \in S} p_s \mathcal{W}_2^2(\nu_{f^*|s}, \nu) = \min_{\mathcal{U}(f)=0} \mathbb{E}\|f^*(X,S) - f(X,S)\|^2.$$

*Thanks to Eq. 20, we can also have*

$$\mathbb{E}\|f^*(X,S) - f_0(X,S)\|^2 = \mathbb{E}\|f^*(X,S) - \bar{g}(X,S)\|^2$$

*and since $f_0$ is fair we can put $\bar{g} = f_0$. This proof is concluded.*

## A.2 Proof of Proposition 2

**Proposition 2** Suppose $K(z) = \frac{1}{\sqrt{2\pi}} \exp(-z^2)$. Conditioning on data $D = \{(x_i, s_i)\}_{i=1}^n$, it holds that

$$\mathcal{U}(\bar{f}(X,S)) \leq \sum_{s \in \mathcal{S}} O\left(p_s n_s^2 \exp\left(-\frac{1}{h^2}\right)\right) + \sum_{s \in \mathcal{S}} O\left(\frac{p_s}{h^4} \mathcal{W}_2^2(\nu_{f^*|s}, \nu_{f^*(D_s)})\right).$$

**Proof** *Recall that $(X,S)$ is the ground-truth distribution. Let*

$$\bar{f}(x,s) := \tilde{T}_{\nu_{f^*(D_s)}, \tilde{\nu}_0}(f^*(x,s)).$$

*Let $\nu_{\bar{f}}$ be the distribution of $\bar{f}(X,S)$, and $\nu_{\bar{f}|s}$ be the distribution of $\bar{f}(X,S)$ conditioning on $S = s$ for $s \in \mathcal{S}$. Suppose $\tilde{\nu}_0$ has $n_0$ supports, denoted by $\{\xi_i\}_{i=1}^{n_0} \subset \mathbb{R}^k$. In the entire proof, we assume $D_s$ is already sampled and thus deterministic for $s \in \mathcal{S}$.*

By (5), we have

$$
\begin{aligned}
\mathcal{U}(\bar{f}) = \min_{\nu} \sum_{s\in\mathcal{S}} p_s \mathcal{W}_2^2(\nu_{\bar{f}|s}, \nu) &\leq \sum_{s\in\mathcal{S}} p_s \mathcal{W}_2^2(\nu_{\bar{f}|s}, \tilde{\nu}_0) \\
&\leq \sum_{s\in\mathcal{S}} 2p_s \mathcal{W}_2^2(\nu_{\bar{f}|s}, \nu_{\bar{f}(D_s)}) + \sum_{s\in\mathcal{S}} 2p_s \mathcal{W}_2^2(\nu_{\bar{f}(D_s)}, \tilde{\nu}_0),
\end{aligned}
\tag{22}
$$

where the second inequality is the triangle inequality. Next, we bound $\mathcal{W}_2^2(\nu_{\bar{f}|s}, \nu_{\bar{f}(D_s)})$ and $\mathcal{W}_2^2(\nu_{\bar{f}(D_s)}, \tilde{\nu}_0)$ from above separately.

Consider any $s \in \mathcal{S}$. We denote the discrete distribution of $\bar{f}(X,S)$ when $(X,S)$ is uniformly sampled from $D_s$ as $\nu_{\bar{f}(D_s)}$. Let $\phi(\cdot) : \mathbb{R}^k \to \mathbb{R}$ be any 1-Lipschitz continuous function on $\mathbb{R}^k$, meaning that $|\phi(\xi) - \phi(\xi')| \leq \|\xi - \xi'\|_2$ for any $\xi$ and $\xi'$ in $\mathbb{R}^k$. We have

$$
\int_{\mathbb{R}^k} \phi(\xi) d\nu_{\bar{f}(D_s)} = \frac{1}{n_s} \sum_{i=1}^{n_s} \mathbb{E}\phi(\tilde{T}_{\nu_{f^*(D_s)}, \tilde{\nu}_0}(f^*(x_i^s, s)) \text{ and } \int_{\mathbb{R}^k} \phi(\xi) d\tilde{\nu}_0 = \frac{1}{n_s} \sum_{i=1}^{n_s} \mathbb{E}\phi(T_{\nu_{f^*(D_s)}, \tilde{\nu}_0}(f^*(x_i^s, s))).
$$

Consider any $i \in \{1, \ldots, n_s\}$. Let $\mathbb{E}_i$ be the conditional expectation conditioning on the outcome of random mapping $T_{\nu_{f^*(D_s)}, \tilde{\nu}_0}(f^*(x_i^s, s)$. By (14) and the 1-Lipschitz continuity of $\phi(\cdot)$, we have

$$
\begin{aligned}
&\left| \mathbb{E}\phi(\tilde{T}_{\nu_{f^*(D_s)}, \tilde{\nu}_0}(f^*(x_i^s, s)) - \mathbb{E}\phi(T_{\nu_{f^*(D_s)}, \tilde{\nu}_0}(f^*(x_i^s, s))) \right| \\
&\leq \left| \mathbb{E}\left[ \mathbb{E}_i \phi(\tilde{T}_{\nu_{f^*(D_s)}, \tilde{\nu}_0}(f^*(x_i^s, s)) - \mathbb{E}_i \phi(T_{\nu_{f^*(D_s)}, \tilde{\nu}_0}(f^*(x_i^s, s))) \right] \right| \\
&\leq \frac{\sum_{j=1}^{n_s} K\left((f^*(x_i^s, s) - f^*(x_j^s, s))/h\right) \mathbb{E}\mathbb{E}_i \left| T_{\nu_{f^*(D_s)}, \tilde{\nu}_0}(f^*(x_i^s, s)) - T_{\nu_{f^*(D_s)}, \tilde{\nu}_0}(f^*(x_j^s, s)) \right|}{\sum_{j=1}^{n_s} K\left((f^*(x_i^s, s) - f^*(x_j^s, s))/h\right)} \\
&= \frac{\sum_{j=1, j\neq i}^{n_s} K\left((f^*(x_i^s, s) - f^*(x_j^s, s))/h\right) \mathbb{E}\mathbb{E}_i \left| T_{\nu_{f^*(D_s)}, \tilde{\nu}_0}(f^*(x_i^s, s)) - T_{\nu_{f^*(D_s)}, \tilde{\nu}_0}(f^*(x_j^s, s)) \right|}{\sum_{j=1}^{n_s} K\left((f^*(x_i^s, s) - f^*(x_j^s, s))/h\right)} \\
&\leq n_s \exp\left( -\frac{\Delta_{f^*(D_s),\min}^2}{2h^2} \right) \Delta_{\tilde{\nu}_0,\max},
\end{aligned}
$$

where

$$
\Delta_{f^*(D_s),\min} := \min_{i,j=1,\ldots,n_s, i\neq j} \|f^*(x_i^s, s) - f^*(x_j^s, s)\| \text{ and } \Delta_{\tilde{\nu}_0,\max} := \max_{j,j'=1,\ldots,n_0} \|\xi_j - \xi_{j'}\|.
$$

As a result, we have

$$
\int_{\mathbb{R}^k} \phi(\xi) d\nu_{\bar{f}(D_s)} - \int_{\mathbb{R}^k} \phi(\xi) d\tilde{\nu}_0 \leq n_s \exp\left( -\frac{\Delta_{f^*(D_s),\min}^2}{2h^2} \right) \Delta_{\tilde{\nu}_0,\max}
$$

for any $\phi$ that is 1-Lipschitz continuous function on $\mathbb{R}^k$. By Kantorovich-Rubinstein duality, we have

$$
\mathcal{W}_2^2(\nu_{\bar{f}(D_s)}, \tilde{\nu}_0) \leq n_s^2 \exp\left( -\frac{\Delta_{f^*(D_s),\min}^2}{h^2} \right) \Delta_{\tilde{\nu}_0,\max}^2 = O\left( n_s^2 \exp\left( -\frac{1}{h^2} \right) \right).
\tag{23}
$$

Again, let $\phi(\cdot) : \mathbb{R}^k \to \mathbb{R}$ be any 1-Lipschitz continuous function on $\mathbb{R}^k$. Let $\mathbb{E}_T$ be the expectation over the random mappings $T_{\nu_{f^*(D_s)}, \tilde{\nu}_0}(f^*(x_i^s, s))$ for $i = 1, \ldots, n_s$. We have

$$
\int_{\mathbb{R}^k} \phi(\xi) d\nu_{\bar{f}(D_s)} = \int_{\mathbb{R}^k} \mathbb{E}_T \Phi(\xi) d\nu_{f^*(D_s)} \text{ and } \int_{\mathbb{R}^k} \phi(\xi) d\nu_{\bar{f}|s} = \int_{\mathbb{R}^k} \mathbb{E}_T \Phi(\xi) d\nu_{f^*|s},
$$

where

$$
\Phi(\xi) = \phi\left( \frac{\sum_{i=1}^{n_s} K\left( \frac{\xi - f^*(x_i^s, s)}{h} \right) T_{\nu_{f^*(D_s)}, \tilde{\nu}_0}(f^*(x_i^s, s))}{\sum_{i=1}^{n_s} K\left( \frac{\xi - f^*(x_i^s, s)}{h} \right)} \right)
$$

*is a random value function whose randomness is from $T_{\nu_{f^*(D_s)}, \tilde{\nu}_0}(f^*(x_i^s, s))$ for $i = 1, \ldots, n_s$. Since $K$ is $O(\frac{1}{h^2})$-Lipschitz continuous and $\phi$ is 1-Lipschitz continuous, $\Phi(\xi)$ is $O(\frac{1}{h^2})$-Lipschitz continuous in $\xi$ given any outcomes of random mappings $T_{\nu_{f^*(D_s)}, \tilde{\nu}_0}(f^*(x_i^s, s))$ for $i = 1, \ldots, n_s$. Therefore, $\mathbb{E}_T \Phi(\xi)$ is also $O(\frac{1}{h})$-Lipschitz continuous.*

*By Kantorovich-Rubinstein inequality (Edwards, 2011), there exists a constant $C$ such that*

$$\int_{\mathbb{R}^k} \phi(\xi) d\nu_{\bar{f}(D_s)} - \int_{\mathbb{R}^k} \phi(\xi) d\nu_{\bar{f}|s} \leq \int_{\mathbb{R}^k} \mathbb{E}_T \Phi(\xi) d\nu_{f^*(D_s)} - \int_{\mathbb{R}^k} \mathbb{E}_T \Phi(\xi) d\nu_{f^*|s} \leq \frac{C}{h^2} \mathcal{W}_2(\nu_{f^*|s}, \nu_{f^*(D_s)}).$$

*which implies*

$$\mathcal{W}_2^2(\nu_{\bar{f}(D_s)}, \tilde{\nu}_0) \leq \frac{C^2}{h^4} \mathcal{W}_2^2(\nu_{f^*|s}, \nu_{f^*(D_s)}). \tag{24}$$

*Applying (23) and (24) to (22) gives the first conclusion.*

*Consider any two groups $s$ and $s'$ in $\mathcal{S}$. Applying (23) and (24) and the triangle inequality again, we have*

$$\begin{aligned}
\mathcal{W}_2^2(\nu_{\bar{f}|s}, \nu_{\bar{f}|s'}) \leq &\mathcal{W}_2^2(\nu_{\bar{f}|s}, \tilde{\nu}_0) + \mathcal{W}_2^2(\nu_{\bar{f}|s'}, \tilde{\nu}_0) \\
\leq &\mathcal{W}_2^2(\nu_{\bar{f}|s}, \nu_{\bar{f}(D_s)}) + \mathcal{W}_2^2(\nu_{\bar{f}(D_s)}, \tilde{\nu}_0) + \mathcal{W}_2^2(\nu_{\bar{f}|s'}, \tilde{\nu}_0) + \mathcal{W}_2^2(\nu_{\bar{f}(D_{s'})}, \tilde{\nu}_0) \\
\leq &O\left((n_s^2 + n_{s'}^2) \exp\left(-\frac{1}{h^2}\right)\right) + O\left(\frac{1}{h^4} \mathcal{W}_2^2(\nu_{f^*|s}, \nu_{f^*(D_s)})\right),
\end{aligned}$$

*which gives the second conclusion.*

## B  Details of Datasets

In this section, we provide more details on the datasets we used in the numerical experiments. We include four datasets from various domains, including marketing domain(*Customer* dataset [4]), medical diagnosis(*Chexpert* Dataset (Irvin et al., 2019)), face recognition (*CelebA* dataset (Liu et al., 2015) and *UTKFace* dataset (Zhang & Qi, 2017)).

The *Customer* dataset has 8068 training samples and 2627 testing samples and the task is to classify customers into anonymous customer categories for target marketing. We partition the data into four sensitive groups based on the gender and the marital status of customers: married female, unmarried female, married male, and unmarried male.

The *Chexpert* dataset contains 224,316 training instances and the task is to detect five chest and lung diseases based on X-ray images. Due to the high computational complexity of solving optimal transportation between large datasets, we sample 5% instances from the original training data as the training set and sample another 5% as the testing set. The *CelebA* dataset contains 162,770 training instances and 39,829 testing instances and the task is to detect ten attributes (chosen based on Ramaswamy et al. (2021)) of the person in an image, which are being attractive, having bags under eye, having black hair, having bangs, wearing glasses, having high cheek bones, being smiling, wearing hat, having a slightly open mouth, and have a pointy nose. For the same computational reason, we sample 5% instances from the original training data as the training set and sample 20% from the original testing data as the testing set. For both Chexpert and CelebA datasets, we partition the data into four sensitive groups based on gender and age: young female, old female, young male, and old male. *UTKFace* dataset consists of 23705 face images with five groups in terms of race(i.e., White, Black, Asian, Indian, and Others) and we randomly split it into training and testing (8:2) sets. And the task is to classify gender and age ($25 \leq$ age $\leq 60$, customized by us) based on face images. All the data in *UTKFace* dataset are utilized.

---

[4]https://www.kaggle.com/datasets/kaushiksuresh147/customer-segmentation

