# OpenReview forum: "Multi-Output Distributional Fairness  via Post-Processing"
_TMLR — Accepted by TMLR_

### Review · Reviewer_DeBp · 2024-11-20

**Summary Of Contributions:**

This paper presents a post-processing method to enhance fairness in **multi-output** machine learning models, addressing limitations of existing methods that are typically designed for single-output models and task-specific fairness measures. The proposed approach focuses on achieving distributional parity, a task-agnostic fairness metric, using optimal transport mapping to align model outputs across different groups towards their empirical Wasserstein barycenter. To manage computational complexity, an approximation technique is used for the barycenter computation, and a kernel regression method is introduced to handle out-of-sample data. Empirical evaluations demonstrate the method’s effectiveness in improving fairness on **multi-task/multi-class** classification and representation learning tasks, outperforming current post-processing baselines.

**Audience:**

Yes

**Broader Impact Concerns:**

The authors may add a section: "Broader Impact Statement"

**Claims And Evidence:**

Yes

**Requested Changes:**

For theoretical part:
- The fairness guarantee of an algorithm is beneficial and will make your paper more theoretically sound for the fairness community. Note that this paper is motivated by [1-2], and both of them present fairness guarantee. Therefore, I suggest the authors to incorporate a bound for their fairness definition, i.e., "Multi-output Distributional Parity." If the authors cannot derive the bound, it is also recommended to add a discussion regarding the fairness guarantee.
- Please discuss equalized odds and equal opportunity in the context of this paper.

For empirical part:
- I believe that it is important to investigate the fairness-utility trade-off [3], especially in the context of multi-output problems. Therefore, I suggest the authors to incorporate some analysis regarding Pareto frontiers, like Figure 4 in [2], which is considered an important reference of this paper.
- Please make a comparison with other fairness-aware algorithms for multi-output problems. See the Weaknesses part.

[1] Fair Regression with Wasserstein Barycenters. NeurIPS 2020.

[2] A minimax framework for quantifying risk-fairness trade-off in regression. The Annals of Statistics 2022.

[3] FACT: A Diagnostic for Group Fairness Trade-offs. ICML 2020.

**Strengths And Weaknesses:**

Strengths:
- The method generalizes fairness notions like Statistical Parity from single-output to **multi-output problems**. To my best knowledge, the motivation is strong since there is a lack of papers discussing multi-output problems in the fair ML community, especially **post-processing**.
- The paper makes **a good combination of previous work to solve multi-output problems**, such as
  - post-processing with fairness constraint defined through optimal transport,
  - the optimal solution as well as its proof technique,
  - the approximation to compute the optimal transport,
  - and the Nadaraya-Watson kernel regression method.
- The numerical experiments demonstrate the method’s superiority compared to existing post-processing techniques, validating its practical utility.

Weaknesses:
- Since the main problem setting, definition, solution, proof and algorithms are strongly motivated by previous work (as the authors state), I would suggest the authors to incorporate more analysis, which are interesting and already presented in these papers. In particular,
  - The fairness guarantees from Proposition 4.1 in [1] and Theorem 7.2 in [2].
  - The analysis of Pareto frontiers in [2].
- In the experimental parts, the paper states that: "Due to the limited post-processing methods in the literature for multi-output problems, we also incorporate an in-processing method, Adversarial Debiasing (Adv Debiasing) (Zhang et al., 2018), which can be **extended** for postprocessing." However, you may not need to "extend" Adversarial Debiasing since both **post-processing** and **multi-output problems** are important advantages that make your paper different from others. Therefore, you can compare your algorithm with **some pre/in-processing methods for multi-output problems** (such as [3-4]) to make your paper more influencial.
- Equalized odds and equal opportunity [5] is also a popular fairness definition in this community. The researchers may naturally raise the question: Can this algorithm be extended to equalized odds and equal opportunity? The authors are encouraged to discuss this fairness definition regarding the proposed algorithm and theoretical analysis.

[1] Fair Regression with Wasserstein Barycenters. NeurIPS 2020.

[2] A minimax framework for quantifying risk-fairness trade-off in regression. The Annals of Statistics 2022.

[3] SimFair: A Unified Framework for Fairness-Aware Multi-Label Classification. AAAI 2023.

[4] Beyond Adult and COMPAS: Fairness in Multi-Class Prediction. arXiv preprint arXiv:2206.07801, 2022.

[5] Equality of Opportunity in Supervised Learning. NeurIPS 2016.

---

> ### Author Response · Authors · 2025-01-20
>
> We thank the reviewer for providing a comprehensive review and helpful comments on our paper. Below, we would like to answer the questions raised.
>
> **Q1:**  I would suggest the authors incorporate more analysis. In particular, (1) a bound for their fairness definition, i.e., "Multi-output Distributional Parity" and (2) some analysis regarding Pareto frontiers.
>
> **A:** In the revised version we uploaded, we have updated Proposition 2 to provide theoretical fairness guarantees analogous to Proposition 4.1 in [1] and Theorem 7.2 in [2]. Specifically, the proposition demonstrates that after applying our post-processing methods, the violation of distributional parity, i.e., $\mathcal{U}(\bar f(X,S))$, converges to zero as the bandwidth $h$ approaches zero and the data sizes $n_s$'s go to infinity. Furthermore, the proposition establishes a similar bound for the Wasserstein distance between the distributions of $\bar f(X,S)$ across any two groups, which corresponds to the fairness guarantees in Proposition 4.1 in [1] and Theorem 7.2 in [2].
>
> The analysis of Pareto frontiers in [2] actually holds exactly for our setting with the same proof. Because of that, we only discuss it in the Remark below Proposition 1.
>
>
> **Q2:** Please make a comparison with other fairness-aware algorithms for multi-output problems to make your paper more influential.
>
> **A:** We thank the reviewer for the constructive suggestion for improving our paper. In our new version, we included three more baselines to further verify the effectiveness of the proposed method. Specifically, we included [3] and [4]for multi-class classification, [4] and [5] for multi-label classification. We can still clearly see the superiority of the proposed method over the baselines. Please refer to our new version of paper for more discussion.
>
>
>
>
> **Q3:** Can this algorithm be extended to equalized odds and equal opportunity? The authors are encouraged to discuss this fairness definition regarding the proposed algorithm and theoretical analysis.
>
> **A:** Yes. It's easy to extend our notion to equal opportunity or equal odds by defining the distributional parity on the conditional distribution of $f(X,S)$, conditioning on the target variable $Y$. Take multi-class classification as an example. Suppose the target is a class label $Y\in\mathcal {Y}$. The extension of equal odds using our notion defines a fairness criterion stating that **$f(X,S)$ conditioning on $S=s$ and $Y=y$ and $f(X,S)$ conditioning on $S=s'$ and $Y=y$ must have the same distribution for any $s,s' \in \mathcal S$ and any $y\in\mathcal {Y}$.** A similar extension can be made for multi-label classification and equal opportunity.
>
> Besides the fairness notion, the proposed post-processing method can also be adapted accordingly for equal odds and equal opporunity. For instance, to achieve equal odds as defined above for multi-class classification, one would first partition the training data into subsets based on the class label $Y$. Then, our method can be applied to each subset to construct a post-processing mapping for each class $Y\in\mathcal{Y}$. For out-of-sample application, one can just  first predict $Y$ using the original (potentially unfair) model and then apply the post-processing mapping corresponding to the predicted $Y$ to achieve fairness within each class."
>
> We have added a remark after Defintion 1 to discuss the extension of the fairness notions and added a discussion on how to extend the algorithm at the end of Section 5.
>
> [1] Fair Regression with Wasserstein Barycenters. NeurIPS 2020.
>
> [2] A minimax framework for quantifying risk-fairness trade-off in regression. The Annals of Statistics 2022.
>
> [3] Baharlouei, Sina, Shivam Patel, and Meisam Razaviyayn. "f-FERM: A Scalable Framework for Robust Fair Empirical Risk Minimization." In The Twelfth International Conference on Learning Representations.
>
> [4] Tifrea, Alexandru, Preethi Lahoti, Ben Packer, Yoni Halpern, Ahmad Beirami, and Flavien Prost. "FRAPPÉ: A Group Fairness Framework for Post-Processing Everything." In Forty-first International Conference on Machine Learning.
>
> [5] Liu, Tianci, Haoyu Wang, Yaqing Wang, Xiaoqian Wang, Lu Su, and Jing Gao. "SimFair: a unified framework for fairness-aware multi-label classification." In Proceedings of the AAAI Conference on Artificial Intelligence, vol. 37, no. 12, pp. 14338-14346. 2023.
>
> **Requested Changes and Broader Impact Concerns**
>
> **A:** We have made the requested changes. Please see our answers to each question above and refer to our new revision.

---

> > ### Comment · Reviewer_DeBp · 2025-01-21
> > **Thanks!**
> >
> > Thanks for your effort! My concern has been properly addressed.

---

### Review · Reviewer_F5TY · 2024-12-10

**Summary Of Contributions:**

This paper introduces a post-processing method for achieving fairness in multi-output machine learning models. It focuses on enhancing distributional parity, which is a task-agnostic measure. The method uses optimal transport to align model outputs across different groups with their empirical Wasserstein barycenter, while maintaining predictive performance. Experimental results on multi-labal/multi-class classification and representation learning tasks demonstrate its effectiveness compared to baseline methods.

**Audience:**

Yes

**Broader Impact Concerns:**

There are no broader impact concerns.

**Claims And Evidence:**

Yes

**Requested Changes:**

The work would be significantly strengthened by incorporating a broader range of baseline methods for comparison. While the authors compare their approach to existing post-processing methods and one in-processing method, the inclusion of additional baselines would provide a more comprehensive evaluation of the proposed method's relative strengths and weaknesses. This would also help clarify the scenarios where the method offers the most advantage.

**Strengths And Weaknesses:**

**Strengths**

1. Extends fairness from single-output models to multi-output settings, addressing an important gap in the literature.

2. Applicable to diverse tasks and models without retraining, offering flexibility and computational efficiency.

**Weaknesses**

1. The method focuses on only one fairness measure, and it is a strong fairness notion. This means that there are cases where the focus on achieving strong distributional parity may lead to large drops in predictive performance.

2. There are limited baselines because the authors choose to primarily target post-processing methods. It would be helpful to have insights on how it compares against pre and in-processing methods.

---

> ### Author Response · Authors · 2025-01-20
>
> Thank you for your constructive comments and feedback on our paper.
>
> **Q1:**  There are cases where the focus on achieving strong distributional parity may lead to large drops in predictive performance.
>
> **A:** We agree with the reviewer that distributional parity is a strong fairness notion, which may result in a greater reduction in predictive performance if fully enforced. However, our method provides flexibility by allowing users to control the extent to which distributional parity is satisfied through the parameter $\alpha\in[0,1]$. This enables users to adjust the fairness level based on the acceptable trade-off with predictive performance.
>
> More importantly, our numerical experiments (see Figures 2, 3(d), and 4(d)) demonstrate that our method does not compromise predictive performance more than other methods across various levels of fairness. Finally, we'd like to emphasize the unique value of the proposed method, given the limited research on multi-output post-processing techniques, particularly in the context of fair representation learning.
>
>
>
> **Q2:** Including additional baselines would provide a more comprehensive evaluation of the proposed method's relative strengths and weaknesses.
>
> **A:** Thank you for your valuable suggestion to improve our paper. In the revised version, we have included three additional baselines to further validate the effectiveness of the proposed method. Specifically, we added [1] and [2] for multi-class classification, and [2] and [3] for multi-label classification. Among these, [2] is a post-processing method, while [1] and [3] are in-processing methods. The results clearly demonstrate the superiority of our proposed method compared to the baselines. For more details, please refer to the updated version of our paper.
>
>
> [1] Baharlouei, Sina, Shivam Patel, and Meisam Razaviyayn. "f-FERM: A Scalable Framework for Robust Fair Empirical Risk Minimization." In The Twelfth International Conference on Learning Representations.
>
> [2] Tifrea, Alexandru, Preethi Lahoti, Ben Packer, Yoni Halpern, Ahmad Beirami, and Flavien Prost. "FRAPPÉ: A Group Fairness Framework for Post-Processing Everything." In Forty-first International Conference on Machine Learning.
>
> [3] Liu, Tianci, Haoyu Wang, Yaqing Wang, Xiaoqian Wang, Lu Su, and Jing Gao. "SimFair: a unified framework for fairness-aware multi-label classification." In Proceedings of the AAAI Conference on Artificial Intelligence, vol. 37, no. 12, pp. 14338-14346. 2023.

---

### Review · Reviewer_NBty · 2025-01-06

**Summary Of Contributions:**

The paper proposes a fair supervised learning framework for multi output/multi class problems under the statistical parity notion. To do so, the paper considers the  weighted distance of every joint distribution of predictions and sensitive attributes to their common barycenter w.r.t. the Wasserstein-2 distance. The main contribution is to generalize the formulation proposed in [1] to multiple classes and multi-tasking problems. Furthermore, due to not accessing to the true joint distributions of predictions and sensitive attributes, they come up with an approximation of Barycenter based on $n$ given iid samples. The experiments demonstrate the superiority of the proposed  approach over several baselines in terms of fairness-accuracy tradeoffs on different multi class/multi label datasets.



[1] Evgenii Chzhen and Nicolas Schreuder. A minimax framework for quantifying risk-fairness trade-off in regression. The Annals of Statistics, 50(4):2416–2442, 2022.

**Audience:**

Yes

**Broader Impact Concerns:**

No Concern.

**Claims And Evidence:**

Yes

**Requested Changes:**

Please see the Weaknesses section in the previous section.

**Strengths And Weaknesses:**

Strengths:

1) The paper is written in a clear way. The algorithm and the provided theory is technically correct.


Weaknesses:

1) The generalization to the multiple classes does not seem a challenging issue in the first place. Please consider to explain why it is not straightforward given [1].

2) There are several method in the literature (both in-processing and post-processing) that can handle multiple sensitive attributes. In particular, [2], [3], and [4] propose their methods in a general case where both sensitive attributes and labels can be non-binary. Please consider comparing your approach with other in-processing (optional) and post-processing (mandatory) methods that can handle multiple labels/classes.

3) In paper [1], Section 7, they propose a post-processing fair approach and they present an approximator of the Barycenter for a given $n$ iid samples. How your estimator is different from the one proposed in that paper? It is important to discuss in what scenarios their estimator fails and how your estimator can address those scenarios.

4) The presented fairness measure only works for demographic parity notion. Is that possible to generalize it to other popular notions such as equalized odds and equal opportunity (by conditioning the joint distributions on $y = 1$)?

5) In the experiments (Figure 2 in particular), the fairness is measured by Wasserstein Distance to Barycenter, which is a measure proposed and directly optimized by the paper. A more fair comparison is to use demographic parity violation (the true unfairness measure of interest) as the fairness measure. Besides, unlike the demographic parity violation, it is not clear how good/bad is Wasserstein Distance to Barycenter = 0.2 for instance. Therefore, please consider to measure fairness with more general criteria such as $p$% rule and demographic parity violation.


[2] Baharlouei, Sina, Shivam Patel, and Meisam Razaviyayn. "f-FERM: A Scalable Framework for Robust Fair Empirical Risk Minimization." In The Twelfth International Conference on Learning Representations.

[3] Tifrea, Alexandru, Preethi Lahoti, Ben Packer, Yoni Halpern, Ahmad Beirami, and Flavien Prost. "FRAPPÉ: A Group Fairness Framework for Post-Processing Everything." In Forty-first International Conference on Machine Learning.

[4] Liu, Tianci, Haoyu Wang, Yaqing Wang, Xiaoqian Wang, Lu Su, and Jing Gao. "SimFair: a unified framework for fairness-aware multi-label classification." In Proceedings of the AAAI Conference on Artificial Intelligence, vol. 37, no. 12, pp. 14338-14346. 2023.

---

> ### Author Response · Authors · 2025-01-20
>
> We thank the reviewer for dedicating the time to provide constructive comments.
>
> **Q1:** The generalization to the multiple classes does not seem a challenging issue in the first place. Please consider to explain why it is not straightforward given [1].
>
> **A:** We would like to clarify that the method in [1] is limited to single-output scenarios because it relies on the cumulative distribution function (CDF) and its inverse. However, the inverse of the CDF is not well-defined for multivariate outputs. Moreover, as introduced in figure 1 of the paper, naively extending the algorithm for single output models to each output of multi-output problems individually does not help reduce the unfairness in the joint distribution of the outputs, making the generalization of [1] to the multiple classes to be a challenging issue.
>
> **Q2:** Please consider comparing with more baseline methods that can handle multiple labels/classes.
>
> **A:** In our new version, we included three more baselines. Specifically, we included [2] and [3]for multi-class classification, [3] and [4] for multi-label classification. In Figure 2, we can still clearly see the superiority of the proposed method over the baselines. Please refer to our new version for more discussion.
>
> **Q3:** How is your estimator different from the one proposed in paper [1], Section 7?
>
> **A:** In Section 7 of paper [1], the proposed Barycenter approximator is restricted to single-output scenarios because it relies on the cumulative distribution function (CDF) and its inverse. This approach fails in our case, as the inverse CDF is not defined for multivariate random variables. In contrast, our proposed method is a multi-output mapping that addresses the gap in multi-output scenarios to achieve distributional fairness.
>
> **Q4:**  Is the presented fairness measure possible to generalize to other notions such as equalized odds and equal opportunity?
>
> **A:** Yes. It's easy to extend our notion to equal opportunity or equal odds by defining the distributional parity on the conditional distribution of $f(X,S)$, conditioning on the target variable $Y$. Take multi-class classification as an example. Suppose the target is a class label $Y\in\mathcal {Y}$. The extension of equal odds using our notion defines a fairness criterion stating that **$f(X,S)$ conditioning on $S=s$ and $Y=y$ and $f(X,S)$ conditioning on $S=s'$ and $Y=y$ must have the same distribution for any $s,s' \in \mathcal S$ and any $y\in\mathcal {Y}$.** A similar extension can be made for multi-label classification and equal opportunity.
>
> Besides the fairness notion, the proposed post-processing method can also be adapted accordingly for equal odds and equal opportunity. For instance, to achieve equal odds as defined above for multi-class classification, one would first partition the training data into subsets based on the class label $Y$. Then, our method can be applied to each subset to construct a post-processing mapping for each class $Y\in\mathcal{Y}$. For out-of-sample application, one can just  first predict $Y$ using the original (potentially unfair) model and then apply the post-processing mapping corresponding to the predicted $Y$ to achieve fairness within each class."
>
> We have added a remark after Definition 1 to discuss the extension of the fairness notions and added a discussion on how to extend the algorithm at the end of Section 5.
>
> **Q5:** In Figure 2, the fairness is measured by Wasserstein Distance to Barycenter, which is proposed and directly optimized by the paper. Please consider more general criteria such as p% rule and demographic parity violation.
>
> **A:** We thank the reviewer for the constructive comment. In Figure 2,  we indeed compared the methods in terms of the demographic parity violation as the reviewer said. This is because the method by Xian et al (2023) directly produces the class labels without generating scores, so Wasserstein distance to barycenter is not defined for their method.
>
> For multi-label classification and fair representation learning, we have to compare the methods in their "Wasserstein Distance to Barycenter". We are not able to use p% rule or demographic parity violation because, **for a multi-label problem, a model can be unfair even if it produces zero violation of demographic parity in each label!** Please see our illustration in Figure 1 of the paper. Also, since our method is designed to achieve distributional parity, which is measured by the Wasserstein distance to barycenter, we need to validate the effectiveness of our method through this metric. Using p% rule or demographic parity violation cannot directly measure the effectiveness of our method.
>
> [1] A minimax framework for quantifying risk-fairness trade-off in regression.
>
> [2] f-FERM: A Scalable Framework for Robust Fair Empirical Risk Minimization.
>
> [3] FRAPPÉ: A Group Fairness Framework for Post-Processing Everything.
>
> [4] SimFair: a unified framework for fairness-aware multi-label classification.

---

> > ### Comment · Reviewer_NBty · 2025-01-29
> > **Thanks for Updating the Paper**
> >
> > Thanks for the clarifications and updating the paper's content. I am satisfied with the revised paper and I found the new revision,  a significant paper in algorithmic fairness for multi-label/output problems.

---

### Decision · Action_Editor_ASqW · 2025-02-18

**Recommendation:** Accept as is

**Comment:**

This paper proposes a post-processing method to enhance distributional parity, a task-agnostic fairness measure, in multi-output machine learning models. Extending previous fairness research on single-output models, this work leverages optimal transport mapping to align the model’s output across different groups. Reviewers agree that this work is well-motivated (DeBp), technically solid (NBty), the proposed post-processing method avoids retraining and is flexible (F5TY), and it fills an important gap between single-output and multi-output fairness in machine learning models (F5TY). Meanwhile, reviewers also raise several concerns regarding the contribution in comparison with previous works (NBty, DeBp), model prediction performance (F5TY), additional criteria (NBty), and additional baseline comparisons (DeBp, F5TY). The authors make a good effort in the rebuttal phase, and reviewers agree that the concerns are resolved in the revised version. While reviewer DeBp highlights that the gap between previous single-output problem and the multi-output problem in this work might be small, reviewer DeBp acknowledges that this is a solid study on a previously under-explored problem.

Given the consensus of reviewers, the AE recommends the acceptance of this paper.

**Audience:**

Yes.

**Claims And Evidence:**

Yes.